# Implicit Gaussian process representation of vector fields over arbitrary latent manifolds

**Robert L. Peach\***
University Hospital Würzburg
peach_r@ukw.de

**Matteo Vinao-Carl\*, Nir Grossman, Michael David**
Imperial College London
(* Indicates equal contribution)

**Emma Mallas, David Sharp, Paresh A. Malhotra**
Imperial College London

**Pierre Vandergheynst, Adam Gosztolai**
EPFL
adam.gosztolai@epfl.ch

## Abstract

Gaussian processes (GPs) are popular nonparametric statistical models for learning unknown functions and quantifying the spatiotemporal uncertainty in data. Recent works have extended GPs to model scalar and vector quantities distributed over non-Euclidean domains, including smooth manifolds appearing in numerous fields such as computer vision, dynamical systems, and neuroscience. However, these approaches assume that the manifold underlying the data is known, limiting their practical utility. We introduce RVGP, a generalisation of GPs for learning vector signals over latent Riemannian manifolds. Our method uses positional encoding with eigenfunctions of the connection Laplacian, associated with the tangent bundle, readily derived from common graph-based data approximation. We demonstrate that RVGP possesses global regularity over the manifold, which allows it to super-resolve and inpaint vector fields while preserving singularities. Furthermore, we use RVGP to reconstruct high-density neural dynamics derived from low-density EEG recordings in healthy individuals and Alzheimer's patients. We show that vector field singularities are important disease markers and that their reconstruction leads to a classification accuracy of disease states comparable to high-density recordings. Thus, our method overcomes a significant practical limitation in experimental and clinical applications.

## 1 Introduction

A cornerstone of statistical learning theory is the *manifold assumption*, which posits that high-dimensional datasets are often distributed over low-dimensional smooth manifolds – topological spaces characterised by local Euclidean structure. For instance, images of an object from varying camera angles or diverse renditions of a written letter can all be viewed as samples from a smooth manifold (Tenenbaum, 2000). Further, the common approximation of data by a proximity graph, based on a notion of affinity or similarity between data points, induces a Riemannian structure that is instrumental in geometric learning theories. For example, the analogy between the graph Laplacian matrix and the Laplace-Beltrami operator associated with a Riemannian manifold (Chung, 1997) has been widely exploited in manifold learning (Belkin & Niyogi, 2003; Coifman et al., 2005), shape analysis (Taubin, 1995), graph signal processing (Ortega et al., 2018), discrete geometry (Gosztolai & Arnaudon, 2021), graph neural networks (Defferrard et al., 2016; Kipf & Welling, 2017; Peach et al., 2020) and Gaussian processes (Borovitskiy et al., 2020; 2021).

However, many datasets contain a richer structure comprising a smoothly varying vector field over the manifold. Prime examples are dissipative dynamical systems where, post an initial transient phase (Fefferman et al., 2016). Likewise, in neuroscience, smooth vector fields arise from the firing rate trajectories of neural populations evolving over neural manifolds, which is instrumental in neural information coding (Sussillo & Barak, 2013; Khona & Fiete, 2022; Gardner et al., 2022).

Smooth vector fields are also pertinent in areas like gene expression profiling during development (La Manno et al., 2018) and multireference rotational alignment in cryoelectron microscopy (Singer & Wu, 2012). These applications emphasise the need to generalise current learning paradigms to capture both the manifold structure and its associated vector field.

To address this need, a promising avenue is to consider the Laplace-Beltrami operator as a hierarchy of Laplacians that act on tensor bundles of a manifold with increasing order. The first member of this hierarchy is the Laplace-Beltrami operator, which acts on rank-0 tensors, i.e., scalar signals. Similarly, higher-order signals, including vector fields, have associated Laplacian operators, which can encode their spatial regularity. Among these, the connection Laplacian (Barbero et al., 2022), defined on vector bundles, and the related sheaf Laplacian (Knöppel et al., 2013; Bodnar et al., 2022), which allows the vector spaces on nodes to have different dimensions, are emerging as leading operators in machine learning (Bronstein et al., 2017; Battiloro et al., 2023; Gosztolai et al., 2023). These operators are related to heat diffusion of higher-order signals over manifolds (Singer & Wu, 2012; Sharp et al., 2019) and thus intrinsically encode the signals' smoothness. The connection Laplacian is particularly appealing because it can be constructed, even when the manifold is unknown, from graph-based data descriptions (Singer & Wu, 2012; Budninskiy et al., 2019). We, therefore, asked how one could use this discrete approximation to derive continuous functions that implicitly represent the vector field over the manifold. Such representation could use the global regularity of the vector field to reconstruct intricate vector field structures lost in data sampling.

Gaussian processes – a renowned family of nonparametric stochastic processes – offer an excellent framework for learning implicit functional descriptions of data. While GPs are traditionally defined on Euclidean spaces (Rasmussen & Williams, 2006), several studies have extended them to Riemannian manifolds. However, these studies have either considered scalar signals (Wilson et al., 2021; Mallasto & Feragen, 2018; Mallasto et al., 2020; Borovitskiy et al., 2020; 2021; Jensen et al., 2020) or vector signals Hutchinson et al. (2021) but only in cases where the underlying manifold is known and is analytically tractable, such as spheres and tori. In this work, we generalise GPs to vector fields on arbitrary latent manifolds, which can only be approximated based on local similarities between data points, making them applicable to real-world datasets.

Our contributions are as follows. (i) We generalise GPs to vector-valued data using the connection Laplacian operator, assuming that the data originates from a stationary stochastic process. (ii) We show that the resulting Riemannian manifold vector field GP (RVGP) method encodes the manifold and vector field's smoothness as inductive biases, enabling out-of-sample predictions from sparse or obscured data. (iii) To underscore the practical implications of RVGP, we apply it to electroencephalography (EEG) recordings from both healthy individuals and Alzheimer's disease patients. The global spatial regularity learnt by our method significantly outperforms the state-of-the-art approaches for reconstructing high-density electrical fields from low-density EEG arrays. This enables our method to better resolve vector field singularities and dramatically increase the classification power of disease states. In sum, our work enables a differential geometric formulation of kernel-based operators and demonstrates a direct relevance for fundamental and clinical neuroscience.

## 2 COMPARISON WITH RELATED WORKS

Let us begin by carefully comparing our method to related works in the literature.

**Implicit neural representations (INRs)**   There has been an increasing interest in defining signals implicitly as parametrised functions from an input domain to the space of the signal. In Euclidean spaces, INRs have been a breakthrough in replacing pixel-wise description images or voxel-wise descriptions of 3D shapes by neural networks (Sitzmann et al., 2020; Lipman, 2021; Gosztolai et al., 2021; Koestler et al., 2022; Mildenhall et al., 2020). INRs have also been extended to signals over manifolds by graph Laplacian positional encoding (Grattarola & Vandergheynst, 2022). However, INRs are data-intensive due to the lack of explicit spatial regularisation. Further, they have not been extended to handle vector-valued data.

**Gaussian processes over specific Riemannian manifolds**   Several closely related works in the GP literature have provided various definitions of GPs on Riemannian manifolds. One line of works defined GPs as manifold-valued processes $f : \mathbb{X} \to \mathcal{M}$ (Mallasto & Feragen, 2018; Mallasto et al.,

2020) by using the exponential map of the manifold to perform regression in the tangent space. However, these works require that the manifold $\mathcal{M}$ be known to define the exponential map. More notable are studies which define GPs as scalar-valued functions $f : \mathbb{X} \to \mathbb{R}$. For example, considering data points as samples from a stationary stochastic process, the domain $\mathbb{X}$ can be defined based on positional encoding using eigenfunctions of the Laplace-Beltrami operator (Solin & Särkkä, 2020; Borovitskiy et al., 2020) or the graph Laplacian (Borovitskiy et al., 2021). However, these works cannot be directly applied to vector-valued signals by treating vector entries as scalar channels. This is because these channels are generally not independent but related through the curvature of the manifold. To address this gap, Hutchinson et al. (2021) defined GPs as functions $f : \mathbb{X} \to \mathcal{TM}$ over the tangent bundle $\mathcal{TM}$ by first mapping the manifold isometrically to Euclidean space and then using a multi-input, multi-output GP to learn the projected signal. However, Hutchinson et al. (2021) has focused on cases when the manifold is explicitly known, and its mapping to Euclidean space can be explicitly defined. Here, we are specifically interested in the case where the manifold is *unknown* – a common scenario in many scientific domains. To achieve this, we generalise the works of Borovitskiy et al. (2020; 2021) to vector fields on Riemannian manifolds using the connection Laplacian operator and its eigenvectors as positional encoding. During reviews, we have also become aware of two related works by Robert-Nicoud et al. (2023); Fichera et al. (2024).

**Gaussian processes in neuroscience**   GPs have also been widely used in the neuroscience literature, particularly combined with linear dimensionality reduction to discover latent factors underlying neural dynamics. A popular method is Gaussian Process Factor Analysis (GPFA) (Yu et al., 2009), which defines GPs in the temporal domain and does not encode spatial regularity over ensembles of trajectories as inductive bias. GPFA has been used to define time-warping functions to align neural responses across trials, i.e., individual presentation of a stimulus or task (Duncker & Sahani, 2018). GPFA has been extended to non-Euclidean spaces by simultaneously identifying the latent dynamics and the manifold over which it evolves (Jensen et al., 2020). However, this model is limited to manifolds built from $SO(3)$ groups and requires them to be enumerated to perform Bayesian model selection. We seek a constructive framework requiring no assumption on the manifold topology.

## 3 BACKGROUND

Here, we introduce the function-space view of GPs in Euclidean domains and their stochastic partial differential equation formulation developed for scalar-valued signals, which will then allow us to extend them to vector-valued signals.

### 3.1 GAUSSIAN PROCESSES IN EUCLIDEAN SPACES

A GP is a stochastic process $f : \mathbb{X} \to \mathbb{R}^d$ defined over a set $\mathbb{X}$ such that for any finite set of samples $X = (\boldsymbol{x}_1, \ldots, \boldsymbol{x}_n) \in \mathbb{X}^n$, the random variables $(f(\boldsymbol{x}_1), \ldots, f(\boldsymbol{x}_n)) \in \mathbb{R}^{n \times d}$ are jointly multivariate Gaussian, $\mathcal{N}(\boldsymbol{x}; \boldsymbol{\mu}, \boldsymbol{K})$, with mean vector $\boldsymbol{\mu} = m(X)$ and covariance matrix $\boldsymbol{K} = k(X, X)$. Consequentially, a GP is fully characterised by its mean function $m(\boldsymbol{x}) := \mathbb{E}(f(\boldsymbol{x}))$ and covariance function $k(\boldsymbol{x}, \boldsymbol{x}') := \mathrm{Cov}(f(\boldsymbol{x}), f(\boldsymbol{x}'))$, also known as the kernel, and denoted as $f \sim \mathcal{GP}(m(\boldsymbol{x}), k(\boldsymbol{x}, \boldsymbol{x}'))$. It is typical to assume that $\boldsymbol{m}(\boldsymbol{x}) = 0$, which does not reduce the expressive power of GPs (Rasmussen & Williams, 2006).

One may obtain the best-fit GP to a set of training data points $(X, \boldsymbol{y}) = \{(\boldsymbol{x}_i, \boldsymbol{y}_i) | i = 1, ..., n\}$ by Bayesian linear regression, assuming that the observations $\boldsymbol{y}_i$ differ from the predictions of $f$ by some Gaussian measurement noise, i.e., $\boldsymbol{y}_i = f(\boldsymbol{x}_i) + \epsilon_i$, where $f \sim \mathcal{GP}(0, k)$ and $\epsilon_i \sim \mathcal{N}(0, \sigma_n^2)$ for some standard deviation $\sigma_n$. Then, the distribution of training outputs $\boldsymbol{y}$ and model outputs $\boldsymbol{f}_* := f(\boldsymbol{x}_*)$ at test points $\boldsymbol{x}_*$, is jointly Gaussian, namely

$$\begin{bmatrix} \boldsymbol{y} \\ \boldsymbol{f}_* \end{bmatrix} \sim \mathcal{N}\left(0, \begin{bmatrix} k(X, X) + \sigma_n^2\mathbf{I} & k(X, X_*) \\ k(X_*, X) & k(X_*, X_*) \end{bmatrix}\right). \tag{1}$$

To generate predictions for a test set $X_* = (\boldsymbol{x}_1, \ldots, \boldsymbol{x}_{n^*})$, one can derive the posterior predictive distribution conditioned on the training set (Rasmussen & Williams, 2006), namely $f_* | X_*, X, \mathbf{y} \sim \mathcal{N}(\boldsymbol{\mu}_{|\mathbf{y}}, \boldsymbol{K}_{|\mathbf{y}})$ whose mean vector and covariance matrix are given by the expressions

$$\boldsymbol{\mu}_{|\mathbf{y}}(\boldsymbol{x}_*) = k(X_*, X)(k(X, X) + \sigma_n^2\mathbf{I})^{-1}\boldsymbol{y}, \tag{2}$$

$$\boldsymbol{K}_{|\mathbf{y}}(\boldsymbol{x}_*, \boldsymbol{x}_*) = k(\boldsymbol{x}_*, \boldsymbol{x}_*) - k(X_*, X)(K + \sigma^2\mathbf{I})^{-1}k(X, X_*). \tag{3}$$

The advantage of GPs is that the smoothness of the training set regularises their behaviour, which is controlled by the kernel function. We focus on kernels from the Matérn family, stationary kernels of the form:

$$k_\nu(\boldsymbol{x}, \boldsymbol{x}') \equiv k_\nu(\boldsymbol{x} - \boldsymbol{x}') = \sigma^2 \frac{2^{1-\nu}}{\Gamma(\nu)} \left( \sqrt{2\nu} \frac{||\boldsymbol{x} - \boldsymbol{x}'||}{\kappa} \right) K_\nu \left( \sqrt{2\nu} \frac{||\boldsymbol{x} - \boldsymbol{x}'||}{\kappa} \right) \tag{4}$$

for $\nu < \infty$, where $\Gamma(\nu)$ is the Gamma function and $K_\nu$ is the modified Bessel function of the second kind. Matérn family kernels are favoured due to their interpretable behaviour with respect to their hyperparameters. Specifically, $\sigma, \kappa, \nu$ control the GP's variability, smoothness and mean-squared differentiability. Moreover, the well-known squared exponential kernel, also known as radial basis function $k_\infty(\boldsymbol{x} - \boldsymbol{x}') = \sigma^2 \exp\left(-||\boldsymbol{x} - \boldsymbol{x}'||^2/2\kappa^2\right)$ is obtained in the limit as $\nu \to \infty$.

## 3.2 Scalar-valued GPs on Riemannian manifolds

In addition to their interpretable hyperparameters, Matérn GPs lend themselves to generalisation over non-Euclidean domains. A formulation that will allow extension to the vector-valued case is the one by Whittle (1963) who has shown that in Euclidean domains $\mathbb{X} = \mathbb{R}^d$ Matérn GPs can be viewed as a stationary stochastic process satisfying the stochastic partial differential equation

$$\left( \frac{2\nu}{\kappa^2} - \Delta \right)^{\frac{\nu}{2} + \frac{d}{4}} f = \mathcal{W}, \tag{5}$$

where $\Delta$ is the Laplacian and $\mathcal{W}$ is the Gaussian white noise. Likewise, for $\nu \to \infty$, the limiting GP satisfies $\exp(-\kappa^2 \Delta/4) f = \mathcal{W}$, where the left-hand side has the form of the heat kernel.

As shown by Borovitskiy et al. (2020), Eq. 5 readily allows generalising scalar-valued ($d = 1$) Matérn GPs to compact Riemannian manifolds $\mathbb{X} = \mathcal{M}$ by replacing $\Delta$ by the Laplace-Beltrami operator $\Delta_\mathcal{M}$. The corresponding GPs are defined by $f_\mathcal{M} \sim \mathcal{GP}(0, k_\mathcal{M})$, with kernel

$$k_\mathcal{M}(\boldsymbol{x}, \boldsymbol{x}') = \frac{\sigma^2}{C_\nu} \sum_{i=0}^{\infty} \left( \frac{2\nu}{\kappa^2} + \lambda_i \right)^{-\nu - \frac{d}{2}} f_i(\boldsymbol{x}) f_i(\boldsymbol{x}'), \tag{6}$$

where $(\lambda_i, f_i)$ are the eigenvalue-eigenfunction pairs of $\Delta_\mathcal{M}$ and $C_\nu$ is a normalisation factor.

## 3.3 Scalar-valued GPs on graphs

Analogously, scalar-valued Matérn GPs can be defined on graph domains, $\mathbb{X} = G$ (Borovitskiy et al., 2021) by using the graph Laplacian in place of $\Delta_\mathcal{M}$ as its discrete approximation. Specifically, the graph Laplacian is $\boldsymbol{L} := \boldsymbol{D} - \boldsymbol{W}$ with weighted adjacency matrix $\boldsymbol{W} \in \mathbb{R}^{n \times n}$ of and diagonal node degree matrix $\boldsymbol{D} = \text{diag}(\boldsymbol{W} \boldsymbol{1}^T)$. The graph Laplacian admits a spectral decomposition,

$$\boldsymbol{L} = \boldsymbol{U} \boldsymbol{\Lambda} \boldsymbol{U}^T, \tag{7}$$

where $\boldsymbol{U}$ is the matrix of eigenvectors and $\boldsymbol{\Lambda}$ is the diagonal matrix of eigenvalues and, by the spectral theorem, $\Phi(\boldsymbol{L}) = \boldsymbol{U} \Phi(\boldsymbol{\Lambda}) \boldsymbol{U}^T$ for some function $\Phi : \mathbb{R} \to \mathbb{R}$. Therefore, choosing $\Phi(\lambda) = \left( 2\nu/\kappa^2 + \lambda \right)^{\nu/2}$ obtains the operator on the left-hand side of Eq. 5 up to a scaling factor[1]. Using this, one may analogously write

$$\left( \frac{2\nu}{\kappa^2} - \boldsymbol{L} \right)^{\frac{\nu}{2}} \boldsymbol{f} = \mathcal{W}, \tag{8}$$

for a vector $\boldsymbol{f} \in \mathbb{R}^n$. Thus, analogously to Eq. 6, the scalar-valued GP on a graph becomes $f_G \sim \mathcal{GP}(0, k_G)$, with kernel

$$k_G(p, q) = \sigma^2 \boldsymbol{u}(p) \Phi(\boldsymbol{\Lambda})^{-2} \boldsymbol{u}(q)^T \tag{9}$$

where $\boldsymbol{u}(i), \boldsymbol{u}(j)$ are the columns of $\boldsymbol{U}$ corresponding to nodes $i, j \in V$, respectively.

# 4 Intrinsic representation of vector fields over arbitrary latent manifolds

We may now construct GPs on unknown manifolds and associated tangent bundles.

---

[1]Note the different signs due to the opposite sign convention of the Laplacians.

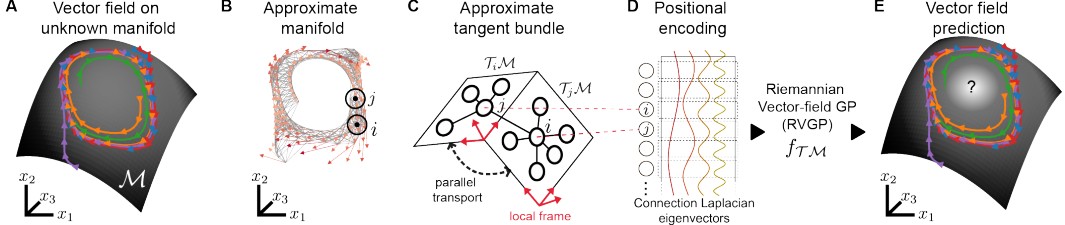

Figure 1: **Construction of vector-valued Gaussian processes on unknown manifolds. A** Input consists of samples from a vector field over a latent manifold $\mathcal{M}$. **B** The manifold is approximated by a proximity graph. Black circles mark two sample points, $i$ and $j$ and their graph neighbourhood. **C** The tangent bundle is a collection of locally Euclidean vector spaces over $\mathcal{M}$. It is approximated by parallel transport maps between local tangent space approximations. **D** The eigenvectors of the connection Laplacian are used as positional encoding to define the GP that learns the vector field. **E** The GP is evaluated as unseen points to predict the smoothest vector field that is consistent with the training data. We use this GP to accurately predict singularities, where sampling is typically sparse.

## 4.1 VECTOR-VALUED GPs ON UNKNOWN MANIFOLDS

We consider training data consisting of pairs $\{(\boldsymbol{x}_i, \boldsymbol{v}_i)|i = 1, \ldots, n\}$, where $\boldsymbol{x}_i$ are samples from a manifold $\mathcal{M} \subset \mathbb{R}^d$ and $\boldsymbol{v}_i$ are sampled from the tangent bundle $\mathcal{TM} = \cup_i \mathcal{T}_i \mathcal{M}$. If the dimension of the manifold is $m \leq d$ the tangent spaces $\mathcal{T}_i \mathcal{M} := \mathcal{T}_{\boldsymbol{x}_i} \mathcal{M}$ anchored to $\boldsymbol{x}_i$ are isomorphic as a vector space to $\mathbb{R}^m$. Importantly, we assume that both $\mathcal{M}$ and $\mathcal{T}_i \mathcal{M}$ are *unknown* and seek a GP to provide an implicit description of the vector field over $\mathcal{M}$ that agrees with the training set and provides a continuous interpolation at out-of-sample test points with controllable smoothness properties.

**Approximating the manifold and the tangent bundle**  We first fit a proximity graph $G = (V, E)$ to $X$, defined based on some notion of similarity (spatial or otherwise) in the data. While $G$ approximates $\mathcal{M}$, it will not restrict the domain to $V$ as in Borovitskiy et al. (2021). Then, to approximate $\mathcal{TM}$, note that the tangent spaces do not have preferred coordinates. However, being isomorphic to $\mathbb{R}^m$, $\mathcal{T}_i \mathcal{M}$ can be parametrised by $m$ orthogonal vectors in the ambient space $\mathbb{R}^d$, to form a local frame, or gauge, $\mathbb{T}_i$. To obtain this frame, we take vectors $\mathbf{e}_{ij} \in \mathbb{R}^d$ from $i$ to $N$ nearest nodes $j$, assuming that they span $\mathcal{T}_i \mathcal{M}^2$ and form a matrix by stacking them column-wise. The left singular vectors corresponding to the $m$ largest singular values yield the desired frame

$$\mathbb{T}_i = (\mathbf{t}_1^{(1)}, \ldots \mathbf{t}_i^{(m)}) \in \mathbb{R}^{d \times m}. \tag{10}$$

Then, $\hat{\boldsymbol{v}}_i = \mathbb{T}_i^T \boldsymbol{v}_i$ acts as a projection of the signal to the tangent space in the $\ell_2$ sense. Note that $m$ does not need to be known ahead of time but is estimated as the average of the dominant single values across all estimated tangent spaces, e.g, based on a cutoff.

**Constraining the vector field over the manifold**  Armed with the approximation of $\mathcal{M}$ and $\mathcal{TM}$, by $G$ and $\{\mathbb{T}_i\}$, we may define the connection Laplacian operator $\boldsymbol{L}_c$ that will regularise the GP's behaviour by using the global smoothness of the vector field.

The notion of vector field smoothness is formalised by the parallel transport map $\mathcal{P}_{j \to i}$ that aligns $\mathcal{T}_j \mathcal{M}$ with $\mathcal{T}_i \mathcal{M}$ to allow the comparison of vectors in a common space. While parallel transport is generally path dependent, we assume that $i, j$ are close enough such that $\mathcal{P}_{j \to i}$ is the unique smallest rotation. Indeed, constructing the nearest neighbour proximity graph limits pairs $i, j$ to be close in space. This is known as the Lévy-Civita connection and is computed as a matrix $\mathbf{O}_{ji} \in O(m)$ in the orthogonal group (rotation and reflection)

$$\mathbf{O}_{ji} = \arg \min_{\mathbf{O} \in O(m)} ||\mathbb{T}_i - \mathbb{T}_j \mathbf{O}||_F, \tag{11}$$

where $|| \cdot ||_F$ is the Frobenius norm and is uniquely computable in $\mathcal{O}(m)$-time (Kabsch, 1976).

---

[2]In practice, we pick $N$ closest nodes to $i$ on the proximity graph where $N$ is a hyperparameter. Larger $N$ increases the overlaps between the nearby tangent spaces. We find that $N = 2D_{ii}$ is often a good compromise between locality and robustness to noise of the tangent space approximation.

Using the parallel transport maps, we define the connection Laplacian (Singer & Wu, 2012), a block matrix $\boldsymbol{L}_c \in \mathbb{R}^{nm \times nm}$, whose $(i, j)$ block entry is given by

$$\boldsymbol{L}_c(i, j) = \begin{cases} D_{ii} \boldsymbol{I}_{m \times m} & \text{for } i = j \\ W_{ij} \boldsymbol{O}_{ij} & \text{for } i, j \text{ adjacent.} \end{cases} \tag{12}$$

Let us remark that $\boldsymbol{L}_c$ prescribes the smoothness of the vector field over an underlying continuous manifold that agrees with the available training data. In fact, as $n \to \infty$, the eigenvectors of $\boldsymbol{L}_c$ converge to the eigenfunctions of the connection Laplacian over the tangent bundle (Singer & Wu, 2017), such that the corresponding continuous signal satisfies the associated vector diffusion process (Berline N., 1996). The solution of this diffusion process minimises the vector Dirichlet energy $\sum_{ij \in E} w_{ij} |\boldsymbol{v}_i - \boldsymbol{O}_{ij} \boldsymbol{v}_j|^2$, which quantifies the smoothness of the vector field (Knöppel et al., 2015).

**Vector-field GP on arbitrary latent manifolds** We can now define a GP to regress the vector field over $\mathcal{M}$. To this end, we consider a positional encoding of points on the tangent bundle $\mathcal{T}\mathcal{M}$ based on the spectrum of the connection Laplacian, $\boldsymbol{L}_c = \boldsymbol{U}_c \boldsymbol{\Lambda}_c \boldsymbol{U}_c^T$, where $\boldsymbol{\Lambda}_c, \boldsymbol{U}_c \in \mathbb{R}^{nm \times nm}$. Compared with Eq. 7, where each node corresponds to a point, each node now represents a vector space of dimension $m$. Thus, the positional encoding of some vector $\boldsymbol{v}$ at node $i$ is given by an $\mathbb{R}^{m \times k}$ matrix, rather than an $\mathbb{R}^{1 \times k}$ vector, whose columns are the eigenvectors corresponding to the $k$ smallest eigenvalues in $\boldsymbol{\Lambda}_c$ and rows are the coordinates of $\mathcal{T}_i \mathcal{M}$:

$$(\widetilde{\boldsymbol{U}}_c)_i = \sqrt{nm} \begin{pmatrix} u_{im,1} & \cdots & u_{im,k} \\ \vdots & & \vdots \\ u_{(i+1)m,1} & \cdots & u_{(i+1)m,k} \end{pmatrix} \in \mathbb{R}^{m \times k}. \tag{13}$$

The matrix $(\widetilde{\boldsymbol{U}}_c)_i$ can also be thought of as the $i$-th slice of an $\mathbb{R}^{n \times m \times k}$ tensor. This allows us to define the positional encoding of $\boldsymbol{v}$ by mapping the eigencoordinates defined in the respective tangent spaces using $\mathbb{T}_i$ back into the ambient space

$$\boldsymbol{P}_{\boldsymbol{v}} = \mathbb{T}_i (\widetilde{\boldsymbol{U}}_c)_i \in \mathbb{R}^{d \times k}. \tag{14}$$

Then, by analogy to Eqs. 5-6, we define the vector-valued Matérn GP $f_{\mathcal{T}\mathcal{M}} : \mathcal{T}\mathcal{M} \to \mathbb{R}^d$ representing the vector field over the manifold with a $\mathbb{R}^{d \times d}$-valued kernel (Fig. 2A, top)

$$k_{\mathcal{T}\mathcal{M}}(\boldsymbol{v}, \boldsymbol{v}') = \sigma^2 \boldsymbol{P}_{\boldsymbol{v}} \Phi(\widetilde{\boldsymbol{\Lambda}}_c)^{-2} \boldsymbol{P}_{\boldsymbol{v}'}^T, \tag{15}$$

where $\boldsymbol{v}, \boldsymbol{v}' \in \mathcal{T}\mathcal{M}$ and $\widetilde{\boldsymbol{\Lambda}}_c = (\boldsymbol{\Lambda}_c)_{1:k, 1:k}$. Note that we recover $f_{\mathcal{M}}$ for scalar signals ($m = 1$), where Laplace-Beltrami operator equals the connection Laplacian for a trivial line bundle on $\mathcal{M}$. Therefore, the tangent spaces become trivial (scalar), recovering the well-known Laplacian eigenmaps $\boldsymbol{P}_{\boldsymbol{x}} = \sqrt{n}(u_{i,1}, \ldots, u_{i,k})$. Likewise, $k_{\mathcal{T}\mathcal{M}}(\boldsymbol{v}, \boldsymbol{v}')$ reduces to the scalar-valued kernel $k_{\mathcal{M}}(\boldsymbol{x}, \boldsymbol{x}')$ for $\boldsymbol{x}, \boldsymbol{x}' \in \mathcal{M}$ in Eq. 6. However, $k_{\mathcal{M}}(\boldsymbol{x}, \boldsymbol{x}')$ and $k_{\mathcal{T}\mathcal{M}}(\boldsymbol{v}, \boldsymbol{v}')$ are not linearly related because the underlying Laplace-Beltrami and connection Laplacian operators are linked by the curvature of the manifold as given by the Weitzenböck identity. Note that the kernel in Eq. 15 differs from that defined in Hutchinson et al. (2021), which considers an isometric projection of the tangent bundle into Euclidean space and constructs a scalar-valued kernel therein (Fig. 2A, bottom). Instead of approximating this isometric projection, which may be challenging for unknown manifolds, our constructive approach uses an intrinsic parametrisation of the manifold using only local similarities. This yields a matrix-valued kernel (Fig. 2A, top), which accounts for alignment of the vector field to principal curvature directions over $\mathcal{M}$. This is made explicit using the connection Laplacian positional encoding (Eq. 13) and tangent spaces (Eq. 10). In Sect. 5, we show that this construction leads to specific performance advantages in recovering vector field singularities.

## 4.2 SCALABLE TRAINING VIA INDUCING POINT METHODS

A drawback of GPs is their computational inefficiency, driven by the need to compute an $n \times n$ covariance matrix. Therefore, inducing point methods, which reduce the effective number of data points to a set of $n' < n$ inducing points, have become a mainstay of GPs in practice (Titsias, 2009). Hutchinson et al. (2021) showed that inducing point methods, e.g., Titsias (2009), are extendible to vector fields over Riemannian manifolds, provided the covariance matrix of inducing points is represented in tangent space coordinates. Since the kernel of RVGP is constructed from a positional encoding using connection Laplacian eigenvectors, which are expressed in local coordinates, our method is readily compatible with inducing point methods, which we provide an implementation of.

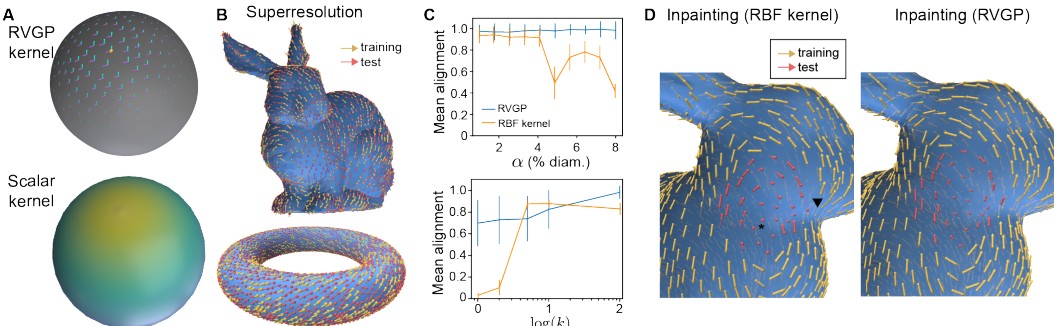

Figure 2: **Superresolution and inpainting.** **A** Matrix-valued kernel (RVGP) against a scalar-valued kernel (e.g., in Hutchinson et al. (2021)). **B** Uniformly distributed samples over the Stanford bunny and torus are interpolated to a higher resolution ($k = 50$). **C** Ablation studies for the Stanford bunny, showing the dependence of alignment of the superresolved vector fields in the test set against data density quantified by the average distance between manifold points $\alpha$ (for $k = 50$ fixed) and the number of eigenvectors $k$. The vectorial RVGP representation is compared against a channel-wise representation using an RBF kernel with Laplacian eigenvectors as positional encoding. **D** Prediction of singularity in masked area. RBF kernel predicts discontinuities along the masked boundary (triangle), vectors that protrude the mesh surface (star) and do not converge to zero magnitude at the singularity. RVGP predicts smoothly varying inpainting ($k = 50$).

## 5 EXPERIMENTS

### 5.1 MANIFOLD-CONSISTENT INTERPOLATION OF VECTOR FIELDS

We expected that RVGP fitted to sparse vector samples over an unknown manifold would leverage the global regularity of the vector field to provide accurate out-of-sample predictions. Thus, we conducted two experiments on the Stanford bunny and toroidal surface mesh to test the RVGP's ability to super-resolve sparse samples and inpaint missing regions containing singularities on diverse manifold topologies. Given $n$ uniformly sampled anchor points $\{x_i\}$ on the surface mesh, we generated a smooth ground truth vector signal over these points by sampling vectors $\{v_i\}$ from a uniform distribution on the sphere $S^2$, projecting them onto their respective tangent spaces $\hat{v}_i = \mathbb{T}_i^T v_i$ and using the vector heat method (Sharp et al., 2019) to find the smoothest vector field. Specifically, concatenating signals as $\hat{v} = \|_{i=0}^{n} \hat{v}_i \in \mathbb{R}^{nm \times 1}$ the vector heat method obtains $\hat{v} \mapsto \hat{v}(\tau)/u(\tau)$, where $\hat{v}(\tau) = \hat{v} \exp(-L_c \tau)$ is the solution of the vector heat equation and $u(\tau) = |\hat{v}| \exp(-L\tau)$ is the solution of the scalar heat equation. We ran the process until diffusion time $\tau = 100$. On surfaces of genus 0 this process will lead to at least one singularity.

**Superresolution** First, we asked if RVGP can smoothly interpolate (also known as super-resolve) the vector field from sparse samples. To this end, we fitted a graph over the bunny mesh and trained RVGP using vectors over $50\%$ of the nodes, holding out the rest for testing. For benchmarking, we also trained a radial basis function (RBF) kernel that treats vector entries as independent scalar channels. We found that RVGP predictions were in excellent visual alignment with the training vector field for dense (Fig. 2B) and sparse data (Fig. S2). The predictions of our model remain accurate when the points do not lie on the manifold surface but are drawn from a distribution centred on the manifold. Indeed, when we added Gaussian geometric noise of increasing standard deviation to manifold points, we found that the prediction accuracy was largely unaffected (Fig. S3).

We conducted two ablation studies to further investigate the robustness of RVGP predictions. First, we subsampled the surface mesh using the furthest point sampling algorithm (Qi et al., 2017) to simultaneously reduce the resolution and the amount of data used for training. Here, a parameter $\alpha$ controls the average pairwise distance of points relative to the manifold's diameter. As quantified by the inner product between predicted and test vectors, RVGP produced accurate alignment with only 10 eigenvectors over a broad range of sampling densities (Fig. 2C). By contrast, the benchmark method suffered a drop in performance at lower densities (higher $\alpha$). Next, we found that on a high-resolution surface ($\alpha = 1.5\%$), RVGP yields high accuracy already with a few eigenvectors

($k$) which progressively increased with $k$. By contrast, the benchmark achieved good representation only from $k = 50$ eigenvectors, thus achieving inferior dimensionality reduction.

**Inpainting** In the second experiment, we tested RVGP's ability to inpaint whole vector field regions containing singularities. This experiment is more challenging than classical inpainting because it requires our method to infer the smoothest topologically consistent vector field. To this end, we masked off the vortex singularity and used the remaining points to train RVGP. We found that vectors predicted by RVGP closely followed the mesh surface, aligned with the training set on the mask boundary and smoothly resolved the singularity by gradually reducing the vector amplitudes to zero (Fig. 2D). By contrast, the RBF kernel did not produce vectors with smooth transitions on the mask boundary and often protruded from the mesh surface, showing that vectors treated as independent scalar fields do not capture the geometry of the tangent spaces. Thus, the connection Laplacian positional encoding provides sufficient regularity to learn vector fields over complex shapes.

## 5.2 Superresolution of EEG data

Finally, as a biologically and clinically relevant use case, we applied RVGP to superresolve electroencephalography (EEG) recordings from humans. EEG recordings measure spatiotemporal wave patterns associated with neural dynamics, which play a fundamental role in human behaviour (Sato et al., 2012; Xu et al., 2023). Accurately resolving these dynamics requires high-density EEG setups in excess of 200 channels (Robinson et al., 2017; Seeber et al., 2019; Siclari et al., 2018). However, due to long setup times during which signal quality can rapidly degrade, experimentalists and clinicians commonly resort to low-density recordings with 32 or 64 channels (Chu, 2015).

Thus, we asked whether superresolving low-density 64-channel recordings (Fig. 3A) using RVGP (Fig. 3C) can facilitate biological discovery and clinical diagnostics. As a ground truth, we collected 256-channel EEG recordings of resting-state brain activity from 33 Alzheimer's patients (AD) and 28 age-matched healthy controls (see Appendix A.1). We focused on low-frequency alpha waves spanning 8-15 Hz, which represent the dominant rhythm at rest (Berger, 1934) and exhibits impaired dynamics in AD (Moretti et al., 2004; Besthorn et al., 1994; Dauwels et al., 2010). After preprocessing of EEG time series (see Appendix A.1), we used a triangulated mesh of the scalp, with 256 known electrode locations as vertices, and finite differencing to compute the corresponding vectors (Fig. 3B,C (ground truth), Appendix A.2). Then, we constructed RVGP kernels using the connection Laplacian eigenvectors derived from a $K$-nearest neighbour graph ($K = 5$) fit to vertices. Finally, we trained RVGP using vectors at 64 training vertices (Fig. 3B) and used it to infer the vectors at the remaining 192 test vertices (Fig. 3C). As a benchmark, we used a channel-wise interpolation of the vector field using linear, spline and RBF kernel methods.

To assess the quality of reconstructions, we computed the divergence and curl of the predicted EEG vector field and computed the mean absolute error (MAE) relative to the ground truth 256-channel EEG. Divergence and curl have previously been used to identify singularities in neural wave patterns in human neuroimaging and have been linked to cognitive function and behaviour (Roberts et al., 2019; Xu et al., 2023). We found that the RVGP reconstruction closely approximated the divergence and curl of the high-density EEG (Fig. 3C). For visual comparison, we show a snapshot with characteristic vector field singularities such as sources, sinks and vortices, which were significantly better preserved by RVGP than benchmarks. Linear interpolation underfitted the vector field, while spline interpolation introduced spurious local structures. RBF kernel interpolation performed most similarly to RVGP due to the homogeneous curvature of the human head. This observation is corroborated by significantly lower angular error and curl-divergence MAE for all subjects for RVGP compared with benchmarks ($n = 61$, Fig. 3D). To test the variability against data density variation, we repeated the experiment for a 32-channel EEG signal and found that the errors were too large for all methods to be of practical relevance (Fig. S1). This shows that 64-channel EEG represents an empirical limit for resolving small-scale singularities.

Given the superior reconstruction accuracy of RVGP, we asked if it could enhance the classification accuracy for patients with Alzheimer's Disease (AD). Contemporary diagnostics for AD are costly, invasive, and laborious (Zetterberg & Bendlin, 2021). We instead employed a linear support vector machine to classify AD patients versus age-matched healthy controls based on the reconstructed divergence and curl fields derived from a brief, one-minute resting-state low-density EEG recording – a procedure that can be feasibly integrated into clinical settings due to its non-invasive nature and

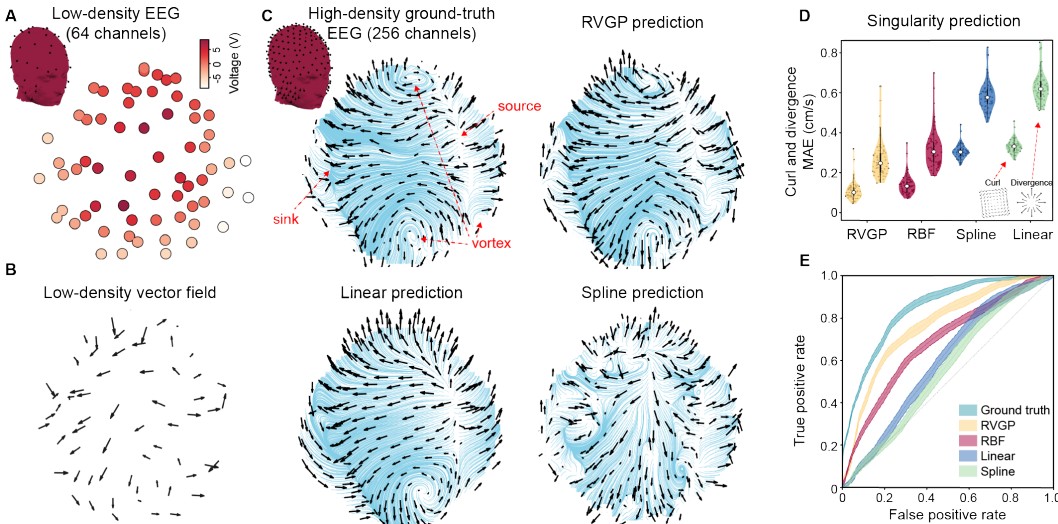

Figure 3: **Reconstruction of spatiotemporal wave patterns in human EEG**. **A** Snapshot of an alpha wave pattern (8-15 Hz) recorded on low density (64 channel) EEG from a healthy subject projected in two dimensions. **B** Phase field of an alpha wave. Vector field denotes the the spatial gradient of the voltage signal. **C** Ground-truth and reconstructed high-density phase field (256 channel) using RVGP, linear and spline interpolation. Streamlines, computed based on the vector field, highlight features of the phase field. RVGP significantly better preserves singularities, i.e., sources, sinks and vortices. **D** Reconstruction accuracy, measured by the preservation of singularities. **E** Receiver operating characteristic (ROC) for binary classification of patients with Alzheimer's disease against healthy controls using a linear support vector machine trained on the divergence and vorticity fields. Shaded areas indicate a 95% confidence interval.

cost-efficiency. Our results show significantly higher than state-of-the-art classification accuracy, approaching that derived from the ground truth high-density EEG signal (Fig. 3E).

# 6 DISCUSSION

We introduced RVGP, a novel extension of Gaussian processes designed to model vector fields on latent Riemannian manifolds. Utilising the spectrum of the connection Laplacian operator, RVGP intrinsically captures the manifold's geometry and topology and the vector field's smoothness. This enables the method to learn global patterns while preserving singularities, filling a significant gap in existing approaches limited to known, analytically tractable manifolds. A key strength of RVGP is its data-driven, intrinsic construction via a proximity graph, which enhances its practical utility by making it highly applicable to real-world datasets where an explicit manifold parametrisation, e.g., for spheres and tori, is often unavailable. Demonstrated across diverse scientific domains such as neuroscience and geometric data analysis, RVGP advances the field of geometrically-informed probabilistic modelling and offers a statistical tool for various high-impact applications, including clinical neuroscience, that is robust to sampling density and noisy data.

As RVGP uses a proximity graph to approximate the manifold, one needs a sampling density that is high enough, combined with suitable similarity metric and graph algorithm to build an intrinsic approximation of the latent manifold, especially in high dimensions. Otherwise, RVGP will use the closest manifold approximated by thresholding the spectral decomposition of the connection Laplacian. Further, although we find that RVGP gives higher reconstruction accuracy than modelling the signal channel-wise on complex manifolds, as expected, this difference diminishes when the manifold curvature is homogeneous. In addition, we considered vector fields that lie in the tangent bundle of the manifold. However, one may also consider non-tangent but smooth vector fields by finding a (non-unique) $SO(d)$ rotation that maps the vectors into the tangent space, applying RVGP, and mapping back the predicted vectors by inverting the rotation. Future work may also study when the tangent spaces are not of uniform dimension by considering the sheaf Laplacian operator.

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

# A EEG ANALYSIS

The following analyses were scripted in MATLAB R2023a:

## A.1 EXPERIMENTAL DATA, AND PRE-PROCESSING

Experimental data: We analysed 10-minute resting state EEG recordings from a cohort of 61 subjects containing 33 patients with a clinical diagnosis of AD and 28 age-matched healthy control subjects. Data was acquired from the clinical imaging facility at Hammersmith Hospital in London. The AD group consisted of 15 Females and 18 Males with a mean age of 76 years (std = 7.8 years), healthy control group contained 15 Females and 13 Males with a mean age of 77 (std = 4.9 years). 10 participants were excluded from the original full cohort of 71 subjects due to poor data quality. During pre-processing, the EEG was bandpass filtered from 2-40 Hz with a second-order Butterworth filter and downsampled to 250Hz. Channels exceeding a kurtosis threshold of 3 were rejected before average re-referencing. We performed an independent component analysis with the Picard algorithm and applied an automatic artefact rejection algorithm in eeglab (Delorme & Makeig, 2004) with a maximum of 5 % of components removed from each dataset. The pre-processed EEG was then filtered in the alpha band (8-15Hz). The low-density EEG consisted of 64 channels sampled evenly across the scalp. These channels were then used to reconstruct the full 256-channel high-density recording.

## A.2 WAVE VELOCITIES

We calculated the velocity vector field at each time point using a method similar to Roberts et al. (2019). We compute the instantaneous phase at each channel using the Hilbert transform and estimate the wave velocity $v$ from the spatial and temporal derivatives of the unwrapped phase $\phi(x, y, z, t)$, as $v = -||\partial t \partial \phi|| / ||\nabla \phi||_2$ implemented using the constrained natural element method (CNEM) (Illoul & Lorong, 2011). CNEM is a mesh-free calculus method for solving partial differential equations that avoids artefacts that can arise from mesh-based interpolation of nodes positioned on the outer boundary of the brain's convex hull.

## A.3 LINEAR AND SPHERICAL SPLINE INTERPOLATION

Spherical spline interpolation was carried out on the raw EEG for each subject using 64 electrodes sampled evenly across the scalp and implemented in eeglab (Delorme & Makeig, 2004). Each channel was mapped onto a unit sphere, and the electrical potential at the missing channel locations was interpolated using the spline function, which weights the contributions of the neighbouring electrode based on its distance to the interpolation point (Perrin et al., 1989).

Linear interpolation was performed on the 64-node 3-dimensional phase flow field. The gradient vector for each node was projected onto the tangent plane prior to linear interpolation.

## A.4 FEATURE EXTRACTION

To determine the behaviourally relevant regions of the cortex, we computed the time-resolved divergence and vorticity fields for each subject. Next, we estimated the probability density of sources, sinks and spirals by computing the mean positive/negative vorticity and mean positive/negative divergence fields thresholded at $> 1$ and $< -1$ averaged over time for each node. This produced four cortical probability maps per subject; mean source probability (positive divergence, $D > 1$), mean sink probability (negative divergence $D < -1$), clockwise spiral probability (positive vorticity, $C > 1$), anticlockwise spiral probability (negative vorticity, $C < -1$). We extracted two predictors from these four probability maps. The first predictor was the probability ratio of sources to sinks at a given node. The second predictor was the probability ratio of clockwise to anti-clockwise spirals at a given node:

$$R_n = \log_{10}\left(\frac{P_n(A)}{P_n(B)}\right) \tag{16}$$

where $P_n(A)$ represents either the probability of a source or a clockwise spiral at node $n$ and $P_n(B)$ represents the probability of a sink or an anti-clockwise at node $n$. We ran a group-level cluster permutation analysis to identify nodes where the source-to-sink ratio or clockwise-to-anti-clockwise spiral ratio differed between the patients with AD and healthy controls in the reference EEG (alpha = 0.05, 50000 permutation). The three clusters with the lowest p-value were then used as seed regions where we extracted the source-to-sink probability ratio and clockwise to anti-clockwise spiral probability ratio in the interpolated EEG for each reconstruction method (RVGP, spline and linear interpolation).

## A.5 BINARY CLASSIFICATION

To evaluate whether the RVGP reconstruction contained relevant information about cognitive function, we tested whether healthy controls and Alzheimer's disease patients could be accurately classified by the ratio of source to sinks and the ratio of clockwise to anti-clockwise spirals. We tested the classification accuracy using three seed brain regions and compared the accuracy from each reconstructed approach (RVGP, spline, linear interpolation) against the high-density reference EEG. For binary classification, we used a linear support vector machine and computed the accuracy and ROC curves for each approach separately using 10-fold cross-validation. We trained two one-vs-all classifiers to categorise pooled samples from all 3 brain regions and across subjects into each class (AD or healthy controls). For each training epoch, we fit the posterior distribution of the scores to 90% of the data and used this to determine the probability of each sample belonging to each class in the test set (10%). To handle class imbalance, we repeated the classification 500 times, with each iteration trained on a random subset of 100 AD and 100 healthy control samples. The AUC and ROC reported in the results were calculated from the distribution estimated over 500 iterations.

## A.6 VISUALISATION

For visualisation, the $x, y, z$ coordinates for each channel location were flattened onto a 2D grid using multi-dimensional scaling to most accurately preserve the local distances between nodes across the scalp. For plotting, we use the 2D tangent vectors of the phase field and streamlines computed using the 'streamslice' function in MATLAB.

## A.7 DATA COLLECTION

Data was collected in the UKDRI Care Research & Technology Centre at the Micheal Uren Hub in London from 2022-2023. AD exclusion criteria were a previous history of head injury or any other neurological condition. All patients had a clinical diagnosis of AD, with one subject retrospectively excluded after a change in diagnosis to frontal-temporal lobe dementia (FTLD). Subjects on AD medication were asked to abstain on the day of testing and to avoid caffeine. 11 subjects were excluded from EEG analysis due to low signal quality, leaving 61 subjects total (33 AD, 28 controls).

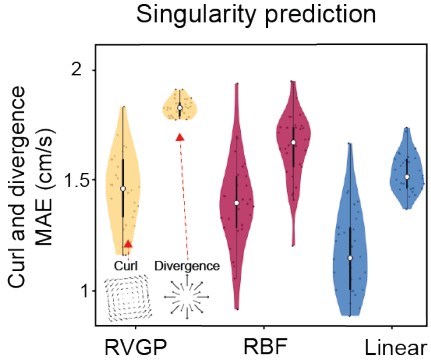

Figure S1: **Reconstruction of spatiotemporal wave patterns in 32-channel human EEG**. Reconstruction accuracy, measured by the preservation of singularities. The accuracy is 10-fold lower than using 64-channel EEG, meaning that predictive power is lost for all methods at this resolution.

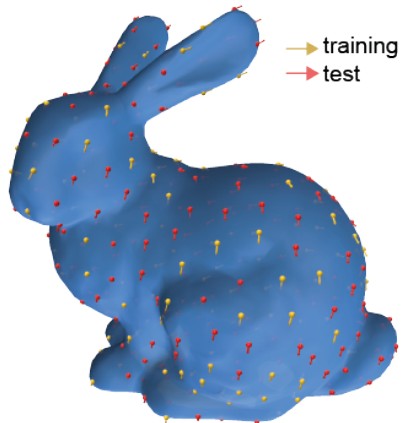

Figure S2: **Superresolution for sparse training data**. Same as in Fig. 2A, but for points placed at an average distance of 5% of manifold diameter.

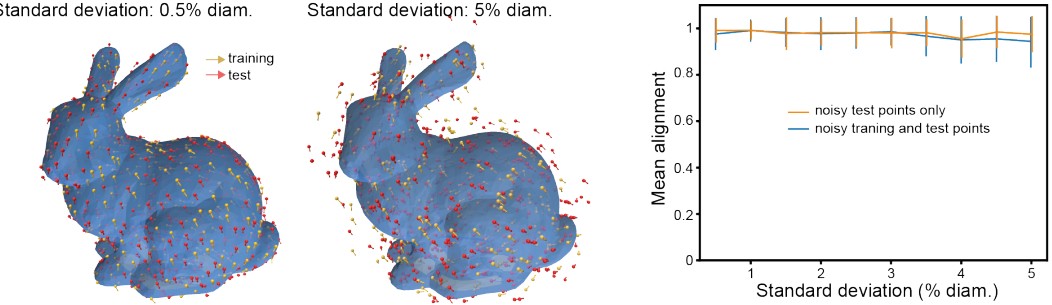

Figure S3: **Training and prediction for off-manifold points**. To test the regularity in our method, we took a distribution of on-manifold points, spaced approximately 3% of manifold diameter, and added additive Gaussian noise to push them off the manifold. We increased the noise until the two dominant dimensions of the tangent space approximations explained at least 80% of the variance, amounting to a standard deviation of approximately 5% of the manifold diameter. We repeated the experiment twice, once with noise affecting only test data points and once adding noise to all data points. We trained the model as described in the main text in Section 5.1. Parameters used: $k = 50$.

