# OpenReview forum: "Implicit Gaussian process representation of vector fields over arbitrary latent manifolds"
_ICLR.cc/2024/Conference — ICLR 2024 poster_

### Official Review · Reviewer_bAaw · 2023-10-28

**Soundness:** 2 fair
**Presentation:** 3 good
**Contribution:** 3 good
**Rating:** 6
**Confidence:** 5

**Summary:**

This paper presents a Riemannian manifold vector field Gaussian process (RVGP), a generalization of Gaussian processes (GPs) for learning vector signals over latent Riemannian manifolds. The core of the idea is to use positional encoding derived from the connection Laplacian. The authors demonstrated the effectiveness of the proposed method via super resolution and inpainting for the vector field on a 3D mesh and EEG analysis.

**Strengths:**

- The theoretical part is mostly well written and technically sound.
- Related work is well addressed.
- Practicality is well demonstrated on real data.

**Weaknesses:**

Lack of precision in some statements.
- V is defined multiply: for vectors and for the nodes of the graph. One of the important statements "While G approximates M it will not restrict the domain to V" becomes unclear.
- The shape of $O_{ij}$ is unclear. It seems to be $m \times m$ for eq. 11, but it seems to be $d \times d$ for eq.12. (The rank of $O_{ij}$ will be m.) The shape of $L_{c}$ is also unclear. It is clearly defined as $nd \times nd$ in eq. 12, but it is inconsistent with $\Lambda_{c},U_{c} \in R^{nm\times nm}$. Maybe something is wrong.
- A quantitative evaluation and analysis is missing for 5.1. I could not judge whether "smoothly resolved the singularity by gradually reducing the vector amplitudes to zero" is okay or not. Does this mean that the result is different from the ground truth? A quantitative evaluation is also appreciated.

Detailed comments:
- Something wrong: {($x_i$}), $\hat{v} = ∥_{i=0}^{n}\hat{v}_{i}$
- I could not understand what the authors were doing: "We then sampled corresponding vectors {$v_{i}$} from a uniform distribution on the sphere".
- $S^{3}$ should be $S^{2}$

**Questions:**

1. The reviewer has a question about the statement "vector field on latent Riemannian manifolds". I suppose two interpretations; each of the vector itself should lie on the manifold, or the domain of the field is enclosed on the manifold but the vector can be out of the manifold. In the experimental result (Fig. 2C), the authors point out the "vectors that protrude the mesh surface". For the former, this should be an undesirable result, but for the latter, it is okay. An example of the latter case is the normal vector of the Stanford bunny for Figure 2. It is also possible to consider such a problem, but I'm curious if the proposed method can model it. At least I expect the proposed method to work for $m=n-1$ (2D manifold in 3D space), but I doubt it for $m<n-1$ because the Levy-Civita connection is insufficient to address the complementary subspace of the tangent space.

2. The reviewer has some doubts about the term "unknown manifold". For 5.1, the training data is dense enough to approximate the manifold well. For 5.2, the authors "constructed RVGP kernels using the connection Laplacian eigenvectors derived from a k-nearest neighbour graph (k = 5) fit to mesh vertices". It seems that the Laplacian was computed from 256 vertices, not n=61. If I understand correctly, this violates the prerequisite of "unknown manifold". Is there no value in analyzing the relationship between the coverage of the data for the manifold and the accuracy of the vector field prediction? I'm also curious about the analysis of the predicted vector field for out of the manifold to check for regularity.

---

> ### Author Response · Authors · 2023-11-17
> **Response to reviewer bAaw part 1**
>
> Dear Reviewer,
>
> We thank you for the positive evaluation of our manuscript and for helping us to improve the mathematical clarity and scope of our method. Please find answers to your concerns below in a three-part message.
>
> Reviewer:
>
> V is defined multiply: for vectors and for the nodes of the graph. One of the important statements "While G approximates M it will not restrict the domain to V" becomes unclear.
>
> Answer:
>
> Thank you for spotting this. We have removed the definition of V relating to vectors as it was unnecessary.
>
> Reviewer:
>
> The shape of O_ij is unclear. It seems to be m x m for eq. 11, but it seems to be d x d for eq.12. (The rank of  will be m.) The shape of L_c is also unclear. It is clearly defined as nd x nd  in eq. 12, but it is inconsistent with \Lambda_c,U_c\in R^{nd \times nd}. Maybe something is wrong.
>
> Answer:
>
> Thank you for spotting this inconsistency. The shape of the matrix Lc is nm x nm, which was a typo. Correspondingly, O_ij is m x m. Both of these operators involve the manifold dimension m because they act intrinsically over the manifold.
>
> Reviewer:
>
> A quantitative evaluation and analysis is missing for 5.1. I could not judge whether "smoothly resolved the singularity by gradually reducing the vector amplitudes to zero" is okay or not. Does this mean that the result is different from the ground truth? A quantitative evaluation is also appreciated.
>
> Answer:
>
> Thank you for requesting further clarifications on this matter. In Fig 2B of the revised version, we now include a quantitative comparison against a method that uses a channel-wise encoding of the vector signal with a radial basis function kernel. We find out our method provides overall higher accuracy and robustness to variation in the data quantity. Furthermore, this accuracy is achieved with fewer eigenvectors, which confirms the efficiency of our positional encoding.
>
> With this additional quantification, Fig 2C now provides a pictorial illustration of the errors stemming from an incomplete satisfaction of the tangency and smoothness constraint by the competing channel-wise RBF method.
>
> Reviewer:
>
> Something wrong: {(x_i}), \hat v = ||_{i=0}^n \hat v_i
> I could not understand what the authors were doing: "We then sampled corresponding vectors {v_j} from a uniform distribution on the sphere".
>
> Answer:
>
> Thank you for spotting the lack of clarity in this paragraph in the previous two bullet points. We have slightly rewritten the paragraph to clarify the construction of the smooth vector field and fixed two minor typos.
>
> Reviewer:
>
> S^3 should be S^2.
>
> Answer:
>
> Thank you, corrected.

---

> > ### Author Response · Authors · 2023-11-17
> > **Response to reviewer bAaw part 2**
> >
> > Reviewer:
> >
> > 1a. The reviewer has a question about the statement "vector field on latent Riemannian manifolds". I suppose two interpretations; each of the vector itself should lie on the manifold, or the domain of the field is enclosed on the manifold but the vector can be out of the manifold. In the experimental result (Fig. 2C), the authors point out the "vectors that protrude the mesh surface". For the former, this should be an undesirable result, but for the latter, it is okay. An example of the latter case is the normal vector of the Stanford bunny for Figure 2. It is also possible to consider such a problem, but I'm curious if the proposed method can model it.
> >
> > Answer:
> >
> > By  "vector field on latent Riemannian manifolds", we mean vectors lying in the tangent bundle associated with the manifold. In Fig 2C, the ground truth vectors are defined to lie in the tangent space of the surface, which is isomorphic as a vector space with R^2. Thus, surface protrusions are undesirable and show that the geometry of the tangent bundle is not approximated by treating vectors channel-wise using Laplacian encoding. Rather, the connection Laplacian operating on vector-valued functions is needed.
> >
> > We clarified this in the text and provided additional quantification of the superiority of our method relative to treating vectors channel-wise and using the RBF kernel (Fig 2B).
> >
> > In principle, our method can model non-tangent vectors over the manifold as long as they vary smoothly. In that case, the way to proceed would be to find a (non-unique) rotation that maps the vector field into tangent vectors, directly apply our method, and invert the rotation. This is necessary because, in RVGP, vectors are first projected to the tangent space coordinates (Equation 10), aligned and operated on in the tangent bundle (Equation 12-13) and then mapped back to ambient space (Equation 14). Because of the projection, information is lost about the orientation and magnitude of vectors out of the tangent space. For example, the normal vector suggested by the reviewer would be projected to the zero vector in tangent space, and after mapping back to ambient space, the vector can have arbitrary magnitude and orientation in or out of the manifold. However, the trick suggested above solves this issue. We mentioned this possibility in the Discussion.
> >
> > Reviewer:
> >
> > 1b. At least I expect the proposed method to work for m=n-1 (2D manifold in 3D space), but I doubt it for m<n-1 because the Levy-Civita connection is insufficient to address the complementary subspace of the tangent space.
> >
> > Answer:
> >
> > Our method is not limited to m=n-1 and certainly applies to the case m<n-1. When the points are closer (by Euclidean norm) than some epsilon threshold over the manifold, the corresponding tangent spaces are also close. In Singer, Wu 2011 (theorem B1, https://arxiv.org/abs/1102.0075), it is shown that epsilon= O(-2/m+2) is good enough, which is the only place where dimension enters the question. Therefore, the Levy-Civita connection is unique. As another consequence, the column spaces of T_i and T_j are also close and are related by the orthogonal transformation O_ij. Because Eq 11 is an overdetermined system, O_ij has a unique solution. Taken together, the discrete approximate connections O_ij are unique, given that the above condition holds.
> >
> > Thus, our method of computing connections can be thought of as a high-dimensional extension of previous methods on meshed surfaces (m=2,n=3) such as https://www.cs.cmu.edu/~kmcrane/Projects/TrivialConnections/.
> >
> > Reviewer:
> >
> > The reviewer has some doubts about the term "unknown manifold". For 5.1, the training data is dense enough to approximate the manifold well.
> >
> > Answer:
> >
> > We thank the Reviewer for bringing up this important point. By ‘unknown manifold’, we mean that the manifold is latent, and we only have access to it through a number of sample points, and there is no a priori parameterization. Our method will implicitly build a manifold irrespective of the sampling density. If the sampling density is too low, the manifold will be an approximation based on the thresholding of the spectral decomposition of the connection Laplacian. We have clarified this in the Discussion.
> > For 5.2, the authors "constructed RVGP kernels using the connection Laplacian eigenvectors derived from a k-nearest neighbour graph (k = 5) fit to mesh vertices". It seems that the Laplacian was computed from 256 vertices, not n=61. If I understand correctly, this violates the prerequisite of "unknown manifold".
> >
> > As the Reviewer correctly points out, in example 5.2, the manifold graph was constructed with the known 256-channel xyz locations, from which our method constructs the underlying latent manifold. The manifold is still unknown because the tangent spaces and connections need to be approximated based on the available data and are not given by an explicit parameterisation (which would exist for canonical manifolds such as the sphere or torus).

---

> > > ### Author Response · Authors · 2023-11-17
> > > **Response to reviewer bAaw part 3**
> > >
> > > Reviewer:
> > >
> > > Is there no value in analyzing the relationship between the coverage of the data for the manifold and the accuracy of the vector field prediction?
> > >
> > > Answer:
> > >
> > > We have decided to use 64-channel EEG as the baseline because it is typical in clinical practice. Our question was: can 64-channel EEG be super-resolved to high-density 256-channel EEG, which is only possible in controlled laboratory settings? We showed that this is indeed possible with RVGP to significantly higher accuracy than previous methods. In the revised manuscript, we now also compare with RBF kernel interpolation, in addition to linear and spline interpolation. This is because while linear and spline interpolation represents the SOTA in EEG analysis practice, Reviewer 1 asked for a stronger baseline, which we thought the RBF kernel would be.
> > >
> > > To address the Reviewer’s concern, we now provide a new analysis of the EEG data with less coverage (32 channels instead of 64). In Fig S1 we show that for 32 channels the errors increase 10-fold for all methods compared with 64 channels. We conclude that 32 channels (12.5% of the data) are too sparse to obtain a high-quality reconstruction, but 64 channels (25% of the data) are enough. The fact that all methods underperform suggests that this has to do with too much discrepancy between the scale of singularities and the sampling resolution. We added these results to the SI with a mention in the main text.
> > >
> > > We also note that this finding is problem-specific, and in Fig. 2B, we show that our method can obtain accurate reconstruction for a range of data densities.
> > >
> > > Reviewer:
> > >
> > > I'm also curious about the analysis of the predicted vector field for out of the manifold to check for regularity.
> > >
> > > Answer:
> > >
> > > Our method, by construction, relies on knowing the anchor points over which the vector field is predicted. Predicting vectors over out-of-sample manifold points would be possible by training another GP, such as the one in Borovitsky et al. 2020 (https://arxiv.org/abs/2010.15538)  to predict the manifold points first. However, in this example, we were interested in predicting the vector field for specific known locations on the human scalp where we have a ground truth signal. Predicting both location and vector field would result in a compound error, which would not fairly represent the accuracy of our method.

---

> > > > ### Comment · Reviewer_bAaw · 2023-11-18
> > > >
> > > > > Reviewer:
> > > > > I'm also curious about the analysis of the predicted vector field for out of the manifold to check for regularity.
> > > >
> > > > > Answer:
> > > > > Our method, by construction, relies on knowing the anchor points over which the vector field is predicted. Predicting vectors over out-of-sample manifold points would be possible by training another GP, such as the one in Borovitsky et al. 2020 (https://arxiv.org/abs/2010.15538) to predict the manifold points first. However, in this example, we were interested in predicting the vector field for specific known locations on the human scalp where we have a ground truth signal. Predicting both location and vector field would result in a compound error, which would not fairly represent the accuracy of our method.
> > > >
> > > > Sorry for the misleading comment.
> > > > The reviewer meant, not specifically for the EEG data, but for a general case as presented in 5.1.

---

> > > > > ### Author Response · Authors · 2023-11-18
> > > > > **Out-of-manifold predictions, Reviewer bAaw**
> > > > >
> > > > > Reviewer: I'm also curious about the analysis of the predicted vector field for out of the manifold to check for regularity.
> > > > >
> > > > > > Apologies, but we assumed that by out-of-manifold, the Reviewer referred to out-of-sample but on-manifold predictions. In example 5.1, we already demonstrated regularity concerning out-of-sample but on-manifold predictions by holding out 50% of data points during training and testing on the other 50% on-manifold points. We do this at varying data densities.
> > > > >
> > > > > > By out-of-manifold predictions, does the Reviewer refer to robustness to noisy samples from the manifold or extrapolation, i.e., points outside the interior or boundary of the manifold? We would appreciate further clarification of the request.

---

> > > > > > ### Comment · Reviewer_bAaw · 2023-11-20
> > > > > > **out-of-manifold prediction**
> > > > > >
> > > > > > The reviewer recognized that the authors already presented  out-of-sample but on-manifold predictions. However, the review had some doubts about the term "unknown manifold" since the training data is dense enough to approximate the manifold well. Therefore, the review requested to check how robust the prediction is for "out-of-manifold", which is NOT on-manifold. For Stanford bunny case, it corresponds the vector field in 3D space, not only on the surface.

---

> > > > > > > ### Author Response · Authors · 2023-11-20
> > > > > > > **Response: out-of manifold prediction, Reviewer bAaw**
> > > > > > >
> > > > > > > Reviewer: The reviewer recognized that the authors already presented out-of-sample but on-manifold predictions. However, the review had some doubts about the term "unknown manifold" since the training data is dense enough to approximate the manifold well.
> > > > > > > Thank you for the additional clarification.
> > > > > > >
> > > > > > > > Before we explain the new experiment and the changes we made, we would like to make sure there is no confusion that by ‘unknown manifold’ we mean that we are given no explicit parametrisation f: M->R^d, in contrast to GP formalisms in previous works. For example, for a sphere, this explicit parametrisation could be a stereographic projection f: S^2->R^2. Rather, our intrinsic parametrisation uses a graph to approximate the manifold and a collection of tangent spaces (Eq. 10) to approximate the tangent bundle.
> > > > > > >
> > > > > > > > To convince the reviewer that our method does not require dense data, in Fig S2 we now include a visual illustration of the Stanford bunny with pairwise distance between points equalling 5% manifold diameter.
> > > > > > >
> > > > > > > Therefore, the review requested to check how robust the prediction is for "out-of-manifold", which is NOT on-manifold. For Stanford bunny case, it corresponds the vector field in 3D space, not only on the surface.
> > > > > > >
> > > > > > > > The core assumption of our method is that the data is distributed on a manifold in R^d. Hence, the question of out-of-manifold needs to be defined carefully. For example, we cannot expect our method to give meaningful predictions for an arbitrary point in R^d far away from the manifold.
> > > > > > >
> > > > > > > > However, we can relax the assumption that the points lie on the manifold surface. Hence, we performed the Stanford bunny experiment in two different ways in the revised version. In the first experiment, we perturbed the manifold points by a Gaussian geometric noise. We increased the noise until the two dominant dimensions of the tangent space approximations explained at least 80% of the variance, amounting to a standard deviation of approximately 5% of the manifold diameter. In the second experiment, we only added noise to the test data while still sampling the training data from the manifold surface.
> > > > > > > In Fig S3, we show that our method displays excellent robustness to geometric noise. We also visually illustrate that the points are indeed significantly perturbed from the manifold surface. These results show that the RVGP method can make accurate predictions for points that are out-of-manifold but drawn from a distribution centred on the manifold.

---

### Official Review · Reviewer_TuVU · 2023-10-30

**Soundness:** 3 good
**Presentation:** 3 good
**Contribution:** 3 good
**Rating:** 8
**Confidence:** 4

**Summary:**

This paper looks at performing Gaussian Process regression over vector fields on unknown manifolds.

The underlying manifold and tangent space is estimated by combining a proximity graph approach to modelling the underlying manifold, and approximating the tangents space by taking the highest singular values of the matrix of directions to neighbours.

The discretised connection Laplacian on these tangent spaces is then used to construct a kernel by projecting the spectral decomposition of the connection laplacian onto the estimated tangent spaces.

This kernel is then used in a number of experiments, from some simple inpainting and super-resolution tasks to superresolution of real EEG data, and show improved diagnostic capabilities using this method.

**Strengths:**

- The method is clean and simple
- The method demonstrably works in the single task presented
- Most of the paper is easy to follow

**Weaknesses:**

- I found the section "Vector-field GP on arbitrary latent manifolds" difficult to follow. For example it is not clear to me what $(U_c)_i$ is. To me this would denote the $i'th$ row, but it clearly is not as it is the wrong shape. Also in equation 15, $\Phi(\Lambda_C)^{-2}$ is a $\mathbb{R}^{mn \times mn}$ matrix, but is being producted with $P_v$, a $\mathbb{R}^{d\times k}$ matrix?

**Questions:**

- Can you explain the procedure of constructing the kernel in "Vector-field GP on arbitrary latent manifolds"?
- How does this kernel differ from using the method of Hutchinson et. al. more directly? I.e. Using the scalar kernel defined by the graph laplacian, $k(i,j)$, from this creating a diagonal kernel $ K(i,j) = k(i,j) * I_{d\times d}$, and then restricting this to the estimated tangent spaces, $\mathbb{K}(i, j) = \mathbb{T}_i K(i,j)  \mathbb{T}_j$
- Presumably one needs to know the dimension of the unknown manifold ahead of time?

---

> ### Author Response · Authors · 2023-11-17
> **Response to reviewer TuVU**
>
> Dear Reviewer,
>
> We thank you for the positive evaluation of our manuscript and for helping us to improve the mathematical clarity and overall comparison of our method to related works. Please find responses to your questions below.
>
> Reviewer:
>
> I found the section "Vector-field GP on arbitrary latent manifolds" difficult to follow. For example it is not clear to me what (U_c)_i is. To me this would denote the ith row, but it clearly is not as it is the wrong shape.
>
> Answer:
>
> We thank the Reviewer for pointing out the lack of clarity in this section. We believe the Reviewer’s confusion is due to points in the tangent bundle representing vector spaces of dimension m rather than points. Therefore, the positional encoding will be a matrix rather than a vector. The index i can be considered a slice of an n x m x k tensor. We have revised the manuscript to clarify this.
>
> Reviewer:
>
> Also in equation 15, \Phi(Lambda_c) is a R^{mn \times mn} matrix, but is being producted with P_v, a R^{d \times k} matrix?
>
> Answer:
>
> Thank you for pointing out this typo. \Phi(Lambda_c) should be \Phi((Lambda_c_{1:k,1:k})) since we are taking the first k eigenvalues.
>
> Reviewer:
>
> Can you explain the procedure of constructing the kernel in "Vector-field GP on arbitrary latent manifolds"?
>
> Answer:
>
> We construct the kernel over the tangent bundle in Equation 15 by direct analogy to the kernel over the manifold in Equation 5-6 by replacing the Laplace Beltrami operator acting on scalar signals on the manifold with the higher-order connection Laplacian operator acting on vector fields over the manifold. This provides a constructive and fully intrinsic definition, i.e., relying on the local Riemannian metric. We will elaborate on this below in the next response.
>
> Reviewer:
>
> How does this kernel differ from using the method of Hutchinson et. al. more directly? I.e. Using the scalar kernel defined by the graph laplacian, k(i,j), from this creating a diagonal kernel K(i,j)=k(i,j)\ast I_{d\times d}, and then restricting this to the estimated tangent spaces, K(i,j) = T_iK(i,j)T_j.
>
> Answer:
>
> Thank you for drawing our attention to a more detailed comparison to Hutchinson et al. First, let us point out that our kernel cannot be obtained by the procedure suggested by the reviewer, i.e., expanding the kernel defined by the graph laplacian, k(i,j), and then restricting to the tangent spaces. This is because the graph Laplacian and the connection Laplacian are not related by linear transformations. Instead, they share a non-linear relationship given by the Weitzenbock inequality, L_c - L = A, where A is an operator depending on the Ricci curvature of the manifold (note that we used Lc and L to define continuous operators obtained as limiting objects of our discrete formulation).
>
> Regarding the comparison to Hutchinson et al., while their construction relies on a non-linear but isometric projection to Euclidean space and defining the GP therein, our method builds the GP intrinsically over the manifold. By intrinsically, we mean using the connection Laplacian operator, which is defined in terms of the Riemannian metric.
>
> Thus, our method is more robust in practice since the connection Laplacian operator can always be constructed from local connections, whereas a global Euclidean isometric projection may be less reliable to approximate.
>
> We have added additional explanations in the text.
>
> Reviewer:
>
> Presumably one needs to know the dimension of the unknown manifold ahead of time?
>
> Answer:
>
> The dimension of the manifold does not need to be known ahead of time. Rather, it can be estimated as the mean of dominant singular values during the tangent space approximation based on a cutoff for the fraction of explained variance. Taking the mean as an estimator is suitable because it minimises the mean-squared error.
>
> We have now implemented this feature in the code (which we will link after publication) and clarified it in the text.

---

> > ### Comment · Reviewer_TuVU · 2023-11-21
> > **Thanks for the clarifications**
> >
> > Thank you for the clarifications - I believe all the parts I mentioned have been clarified satisfactorily.
> >
> > I asked "How does this kernel differ from using the method of Hutchinson et. al. more directly?" as I am sure your method is better and provides a more sensible kernel. It would be very nice I think if it were possible to visualise the kernel.
> >
> > For example, akin to how one plots a scalar kernel as a line/colormap by evaluating k(?, x) over the domain, it is possible to plot the co-vector field (or the associated vector field) defined by k(?, (x, v)) for a chosen point and tangent vector. Compare this to the kernel from Hutchinson et. al. on e.g. the sphere and I guess that you will end up with a more sensible picture, rather than a kernel that depends on the chosen embedding into euclidean space. I think this would be a very good visual sell for the paper.
> >
> > On a related point, does this methodology extend to the continuous setting where one knows the manifold ahead of time, akin  to Borovitskiy et al., 2021? What are the challenges associated?

---

> > > ### Author Response · Authors · 2023-11-22
> > > **Kernel visualisation, Reviewer TuVU**
> > >
> > > I asked "How does this kernel differ from using the method of Hutchinson et. al. more directly?" as I am sure your method is better and provides a more sensible kernel. It would be very nice I think if it were possible to visualise the kernel.
> > >
> > > For example, akin to how one plots a scalar kernel as a line/colormap by evaluating k(?, x) over the domain, it is possible to plot the co-vector field (or the associated vector field) defined by k(?, (x, v)) for a chosen point and tangent vector. Compare this to the kernel from Hutchinson et. al. on e.g. the sphere and I guess that you will end up with a more sensible picture, rather than a kernel that depends on the chosen embedding into euclidean space. I think this would be a very good visual sell for the paper.
> > >
> > > > We thank the Reviewer for this insightful suggestion. In Fig. 2, we now provide the visualisation of our kernel and a comparison to the kernel used in Hutchinson et al. 2021. Note that our kernel is matrix-valued and thus is plotted as three vector fields, only two of which are relevant as they are approximately orthogonal and provide a parametrisation in principal curvature directions. These vector fields diminish radially, acting as a measure of similarity, and vary along the manifold to inform of manifold curvature in principal directions. Contrasting this with the scalar kernel k(i,j) of Hutchinson et al, 2021, which they extend into a vector kernel as K(i,j) = k(i,j)*I_d, one can see that our kernel is not related by a linear transformation to theirs because it also encodes the curvature of the manifold.
> > >
> > > On a related point, does this methodology extend to the continuous setting where one knows the manifold ahead of time, akin to Borovitskiy et al., 2021? What are the challenges associated?
> > >
> > > > This is a very good question. We think our discrete construction approximates a continuous kernel operator, which could be rigorously shown using the methods in Singer, Wu, "Spectral convergence of the connection Laplacian from random samples" (2017). One would need to prove the convergence of these objects as the spacing of uniformly distributed samples diminishes, which seems possible. More broadly, it would also be interesting to study when the samples are not necessarily densely and uniformly distributed. This is precisely the case where a known manifold prior would be useful, as the reviewer suggests. However, this is out of the scope of this work.

---

### Official Review · Reviewer_3KnT · 2023-11-06

**Soundness:** 2 fair
**Presentation:** 3 good
**Contribution:** 3 good
**Rating:** 8
**Confidence:** 2

**Summary:**

Gaussian processes is a popular Bayesian method that easily incorporates prior knowledge and provides good uncertainty quantification. GP is originally defined in Euclidean space which limits its application in certain domains. In this paper, the author proposes the Riemannian manifold vector field GP (RVGP) which extends GP to learn vector signals over latent Riemannian manifolds with the use of connection Laplacian operator. Experiment results show RVGP can encode the manifold and vector field's smoothness as inductive biases and have good performances on electroencephalography recordings in the biological domain.

**Strengths:**

- Extending GP to vector-valued signals is novel. The proposed method also removes the common assumption that the manifold is known in non-Eucleadian GP, which expands GP's applicability.
- The paper is well written.

**Weaknesses:**

- One advantage of GP is data efficiency. In section 5.1's superresolution experiment RVGP is trained using vectors over 50% of the nodes, I would be interested to see the results when trained with less data.
- In section 5.2 the authors compare RVGP with interpolation methods, which might be a too simple baseline comparison. The RVGP's performance is also close to linear prediction. I would be interested to see the comparison against a stronger baseline method.
- A section discussing the limitations of the proposed method would be good.

**Questions:**

See weakness above.

---

> ### Author Response · Authors · 2023-11-17
> **Response to reviewer 3KnT**
>
> Dear Reviewer,
>
> We thank you for the positive evaluation and the criticisms of our article. We have now addressed your concerns and believe the article has been substantially improved. Please find individual responses below.
>
> Reviewer:
>
> One advantage of GP is data efficiency. In section 5.1's superresolution experiment RVGP is trained using vectors over 50% of the nodes, I would be interested to see the results when trained with less data.
>
> Answer:
>
> We thank the Reviewer for highlighting this point. In Fig 2B, we already vary the density of data points, hence the amount of training data, by changing the resolution of the surface representation. We showed that there is no significant drop in accuracy when we vary the pairwise distance of sample points relative to the object (bunny) diameter. Increasing from 1% to 8% effectively uses less data and results in a smoother bunny surface due to the lower resolution.
>
> To address the Reviewer’s concern, we have now added a benchmarking of our method, which is based on a vectorial representation using the connection Laplacian, against an alternative GP method based on a channel-wise representation using the graph Laplacian and a radial basis function kernel. Unlike the benchmark, our method shows no significant variation against resolution and can represent the vector field with significantly fewer eigenvectors (Fig 2B).
>
> Reviewer:
>
> In section 5.2 the authors compare RVGP with interpolation methods, which might be a too simple baseline comparison. The RVGP's performance is also close to linear prediction. I would be interested to see the comparison against a stronger baseline method.
>
> Answer:
>
> Thank you for pushing us to perform further benchmarking of RVGP. We chose linear and spline interpolation as benchmarks since there are broadly adopted approaches in current experimental and clinical practice.
>
> At the request of the reviewer, we now include a comparison of our vector-based method against a channel-wise radial basis function (RBF) kernel. We show in Fig. 3D,E that this method offers higher performance than linear and spline interpolation but is subpar compared to RVGP.
>
> Reviewer:
>
> A section discussing the limitations of the proposed method would be good.
>
> Answer:
>
> In the revised version, we now include an expanded Discussion section detailing the limitations and future possibilities.

---

> > ### Comment · Reviewer_3KnT · 2023-11-22
> >
> > Thank you for the clarifications and additional experiments. It cleared my concerns and I decided to raise my score.

---

### Author Response · Authors · 2023-11-17
**Revision of article and response to reviewer requests**

Dear Reviewers,

Thank you for the positive evaluation of our manuscript.

We have now submitted the revised version, in which we believe we addressed all your concerns. Specifically, we now provide additional benchmarks, further experiments varying the data density, and clarifications and corrections to the exposition of our method as well as its limitations.

Please refer to individual comments to find responses to your questions.

Kind regards,
The authors

---

### Meta-Review · Area_Chair_3XiZ · 2023-12-05

**Metareview:**

This paper was reviewed by three reviewers, who found the proposed method elegant and technically sound, the presentation clear, and the experiments insightful. The paper was subject to active discussion during the rebuttal (author-reviewer discussion) phase, and the updates improved the paper. In the end, all reviewers recommend accepting this work.

**Justification For Why Not Higher Score:**

The topic is specialised and even if I would be happy to see this as a spotlight, I understand that we cannot accept all papers as spotlights.

**Justification For Why Not Lower Score:**

The topic area is slightly esoteric, but I would still rather accept the paper than reject it.

---

### Decision · Program_Chairs · 2024-01-16

Accept (poster)